# Yaw Rate Prediction and Tilting Feedforward Synchronous Control of Narrow Tilting Vehicle Based on RNN

Ruolin Gao [1], Haitao Li [1,*], Ya Wang [2], Shaobing Xu [2,3], Wenjun Wei [1,2], Xiao Zhang [1] and Na Li [1]

1 College of Engineering, China Agricultural University, Beijing 100083, China; grlbest@cau.edu.cn (R.G.)
2 Beijing Zuoqi Technology Co., Ltd., Beijing 100083, China
3 School of Vehicle and Mobility, Tsinghua University, Beijing 100083, China
* Correspondence: h.li@cau.edu.cn; Tel.: +86-132-4096-7299

**Abstract:** The synchronous control of yaw motion and tilting motion is an important problem related to the lateral stability and energy consumption of narrow tilting vehicles. This paper proposes a method for the tilting control of narrow tilting vehicles: tilting feedforward synchronous control. This method utilizes a proposed novel prediction method for yaw rate based on a recurrent neural network. Meanwhile, considering that classical recurrent neural networks can only predict yaw rate at a given time, and that yaw rate prediction generally needs to analyze a large amount of computer vision data, in this paper, the yaw rate is represented by a polynomial operation to predict the continuous yaw rate in the time domain; this prediction is realized using only the driving data of the vehicle itself and does not include the data generated by computer vision. A prototype experiment is provided in this work to prove the advantages and feasibility of the proposed tilting feedforward synchronous control method for narrow tilting vehicles. The proposed tilting feedforward synchronous control method can ensure the synchronous response of the yaw motion and the tilting motion of narrow tilting vehicles.

**Keywords:** narrow tilting vehicles; tilting control; synchronous control; prediction of yaw rate; recurrent neural network

## 1. Introduction

Narrow tilting vehicles (NTVs) are highly maneuverable in their ability to drive quickly in curves and maintain excellent lateral stability [1,2]. The synchronous control of vehicle tilting and yaw is the key to achieving lateral stability [3]. When the active tilting motion of NTVs is synchronized with the yaw motion, the moment of the roll degree of freedom (DoF) is zero, the active tilting motion consumes less energy, and the asymmetric wear of the tires on both sides is reduced [4,5]. However, the active tilting of the NTV to the left and right is achieved by the suspension on the left and right sides moving up and down [6,7], respectively. The suspension of NTVs is inevitably fitted with shock absorbers. The shock absorbers provide some damping to improve the comfort of NTVs on uneven roads [8,9]. These dampers also contribute when the suspension is moved up and down to achieve the active tilting motion of the NTVs. As a result, the response of active tilt motion lags behind yaw motion, rather than synchronous response [4].

Many methods have been proposed to solve this problem. Claveau et al. [10,11] found that controlling the actuators of active tilting motion and yaw motion based on the driver's steering wheel angle feedback can reduce the response lag of the tilting angle and designed a nonlinear control method. Nguyen et al. [5] proposed a linear variable parameter control method based on the feedback of vehicle motion states, such as driving velocity and yaw rate, to reduce the error of the tilting angle in the process of yaw motion. Tang et al. [12–14] studied the NTV dynamic of rollover critical conditions. A roll index based on yaw rate feedback is proposed to calculate the target tilting angle. They designed a model predictive controller based on the rollover index. The controller reduces the error between the true

and target tilting angles, improves the handling performance of the NTV, and ensures the lateral and roll stability of the NTV. Ren [4] proposed a tilting control method based on a torque vector. Based on NTV drive torque feedback and steer-based tilting control [15,16], this method reduces the average relative error of the tilting angle to 1.546%.

The above methods of tilting control are based on the feedback of the driver's operation or the vehicle's motion state [9,17]. These control methods analyze the dynamics of the NTV suspension to speed up the tilting response through the cooperative output of single or multiple actuators [10,11]. Due to the principle of feedback control and the dynamic characteristics of damping, traditional methods can only speed up the tilting response as much as possible [15,16]. In fact, there is always a lag between the active tilting motion and the yaw motion [12–14].

We propose a tilting feedforward synchronous control (TFSC) method. In this method, the yaw motion of the NTV is predicted first, and the predicted value is earlier than the tilting motion in the time domain. The predicted value is used as the feedforward of tilting control, which is more conducive to the synchronization of active tilting motion and yaw motion.

Behavior prediction of dynamic systems such as yaw rate prediction is an important problem in the context of intelligent vehicles [18,19]. Recurrent neural networks (RNN) have been widely used to solve this problem [20]. Many methods have been proposed to predict the motion of vehicles or obstacle vehicles [21,22]. Patel et al. [20] found that the use of RNN to analyze road condition information could predict and classify driver intentions in the next 3 s. Liu et al. [21] proposed a depth algorithm based on RNN to predict vehicle mobility in the next 10–30 min. The input of this method contains a large area of road condition information. Although the prediction result is accurate, the computation time of each step is long. Min et al. [22] proposed a trajectory prediction method for obstacle vehicles based on RNN deep integration. The input of the integrated network model includes the distances of other moving agents on the road. This prediction method could predict the trajectory of obstacle vehicles in the next 2 s.

First, the classical RNN method can only predict the yaw rate at a given time [21]. Then, all of these methods incorporate a large amount of environmental information through computer vision (CV) [23], including moving agents (e.g., pedestrians and vehicles) and road context information (e.g., lanes, traffic lights) [24]. The driving status information of the vehicle itself is rarely considered. For NTVs, as a short urban commuting tool, due to their compact body, high mobility, multi-DoF motion (tilt, yaw, lateral advance, and longitudinal advance), and low cost characteristics, it does not necessarily need to consider complex environmental information but needs to pay attention to the driving state of the NTV itself.

To further boost the computational speed of the prediction algorithm and provide a yaw rate prediction method for NTVs, we propose a prediction algorithm based on RNN. This algorithm can achieve the continuous prediction of vehicle yaw rate by analyzing only the data generated by the vehicle itself (including lateral and longitudinal driving velocity and displacement), avoiding the large amount of data generated by CV.

This work aims to predict the yaw rate of NTVs and realize the synchronous control of the yaw rate and tilting angle based on the prediction. The contributions include:

(1) A calculating method for predicting the yaw rate is proposed. The NTV yaw rate is represented by a polynomial operation to predict the continuous yaw rate in the time domain.

(2) The tilting feedforward synchronous control (TFSC) method for NTVs based on the predicted value of the yaw rate is proposed.

(3) A network model is designed based on RNN to predict the coefficients of the polynomial operation. The model is trained on real driving data collected by an NTV prototype.

(4) The NTV prototype is used to collect vehicle driving data, and the network model works entirely with data obtained from onboard sensors. The feasibility of the TFSC method is verified by the prototype experiment.

The remainder of this paper is organized as follows: The calculating method and the network model for predicting are studied and trained in Section 2. Section 3 presents the TFSC method for NTVs based on the predicted value of the yaw rate. Experiments are studied in Section 4 to compare the TFSC method and the traditional direct tilt control (DTC) without prediction. The discussion and conclusions are presented in Sections 5 and 6, respectively.

## 2. Mathematics and Network Model for Prediction

This section describes how to predict the future yaw rate of an NTV based on RNN. The predictive value is used in the TFSC. The accurate predictive value is the basis for realizing TFSC.

### 2.1. Mathematics, Input, and Output

To predict the continuous yaw rate in the time domain, it is expressed as a time-dependent polynomial operation. The polynomial operation has three coefficients. The operation method is shown in Figure 1.

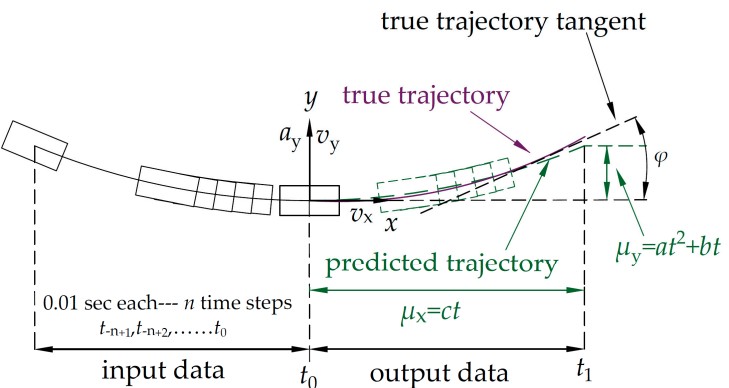

**Figure 1.** The diagram of the algorithm.

The NTV coordinate at time $t_0$ is located at the center of the NTV at time $t_0$, and the x-axis points to the heading of the NTV. In the coordinate, the real trajectory before time $t_0$ is divided into $n$ steps with an interval of 0.01 s. Since the interval of each step is very short in the calculation, the lateral acceleration ($a_y$) and longitudinal velocity ($v_x$) between each step are assumed to be constant. In the NTV coordinate system at time $t_0$, the predicted trajectory of the NTV can be expressed as follows:

$$\begin{cases} \mu_y = at^2 + bt \\ \mu_x = ct \end{cases} \tag{1}$$

where $t \in [t_0, t_1]$; $\mu_x / \mu_y$ is the longitudinal/lateral displacement of the NTV in the time $t$; $a$, $b$, and $c$ are the three coefficients to be calculated.

Therefore, relative to the NTV coordinate at time $t_0$, the yaw angle $\varphi$ at time $t$ can be expressed as the angle between the x-axis of NTV at time $t$ and the x-axis of NTV at time $t_0$, that is, the angle between the trajectory tangent at time $t$ and the x-axis of NTV at time $t_0$. Therefore, the yaw angle $\varphi$ of the predicted trajectory can be expressed as:

$$\varphi = \arctan\left(\frac{2at + b}{c}\right) \tag{2}$$

where $\left(2\frac{a}{c}t + \frac{b}{c}\right)$ is the slope of the trajectory tangent.

Thus, the yaw rate $\dot{\varphi}$ of NTV can be denoted as:

$$\dot{\varphi} = \frac{2ac}{4a^2t^2 + 4abt + b^2 + c^2} \tag{3}$$

The calculated $\dot{\varphi}$ is defined in the NTV coordinate at time $t_0$. The input of the network model is also defined in the NTV coordinate at time $t_0$, including the lateral acceleration $(a_y)$, the longitudinal/lateral velocity $(v_x/v_y)$, and the coefficients $a$, $b$, and $c$ of the $\dot{\varphi}$ before the time $t_0$.

$$input = \begin{bmatrix} a_y & v_x & v_y & a & b & c \end{bmatrix} \tag{4}$$

Since the network model needs to process time-sequential data, the input data of $n$ steps before time $t_0$ form an $n \times 6$ time-sequential matrix with a 0.01 s interval as the input of the network model. The network model gives the output by calculating the input.

The output of the network model is the yaw rate coefficients $a_{pred}$, $b_{pred}$, and $c_{pred}$ after the time $t_0$. According to (3) and the output coefficients, the predicted yaw rate ($\dot{\varphi}_{pred}$) can be determined by (5).

$$\dot{\varphi}_{perd} = \frac{2a_{perd}c_{perd}}{4a_{perd}{}^2 t^2 + 4a_{perd}b_{perd}t + b_{perd}{}^2 + c_{perd}{}^2} \tag{5}$$

### 2.2. Network Model

The network model is designed based on RNN. The input is $n$ time-sequential data points with 6 variables. The interval of the time-sequential data points is 0.01 s. Therefore, the size of the input data is [batch $\times$ $n$ $\times$ 6].

The structure of the network model is shown in Figure 2. Input $t_i$, i $\epsilon$ $(-n+1,0)$ represents the $n$ time-sequential data points inputted in $n$ operation steps. Furthermore, i is an integer.

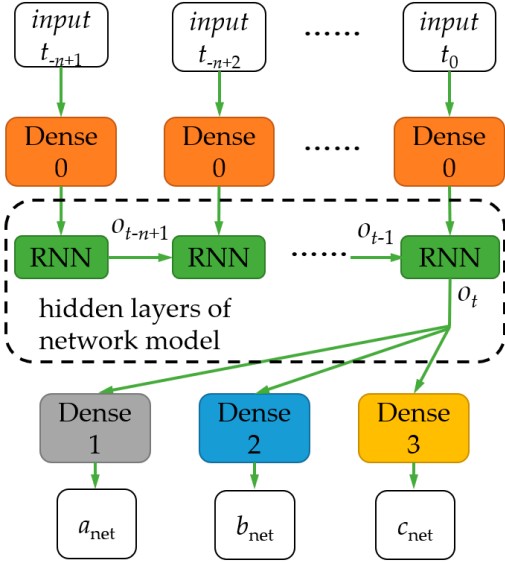

**Figure 2.** Structure of RNN.

Before the input data enter the hidden layer, a dense layer [25] (Dense0) is calculated. The input is converted into a vector of length *Non* after the Dense0 calculation. *Non* is the number of nodes (*Non*) in each hidden layer.

After the hidden layer calculation, the output of the hidden layer $o_t$ is used for three other different dense layer operations (Dense1, Dense2, and Dense3), used to generate 3 coefficients as the output: $a_{net}$, $b_{net}$, and $c_{net}$. The 3 coefficients are obtained by performing linear operations on the output $o_t$ of the last time step.

$$\begin{bmatrix} a_{net} \\ b_{net} \\ c_{net} \end{bmatrix} = \begin{bmatrix} W_a \\ W_b \\ W_c \end{bmatrix} o_t + \begin{bmatrix} b_a \\ b_b \\ b_c \end{bmatrix} \tag{6}$$

where $W_a$ is the weight used to calculate the coefficient $a_{net}$; $b_a$ is the bias of the coefficient $a_{net}$; $W_b$ is the weight used to calculate the coefficient $b_{net}$; $b_b$ is the bias of the coefficient $b_{net}$; $W_c$ is the weight used to calculate the coefficient $c_{net}$; and $b_c$ is the bias of the coefficient $c_{net}$.

The final coefficients for predicting the yaw rate are obtained by dividing $a_{net}$, $b_{net}$, and $c_{net}$ by $\lambda_a$, $\lambda_b$, and $\lambda_c$, respectively. As the weight variables defined for training the network model, $\lambda_a$, $\lambda_b$, and $\lambda_c$ are described in the training section of the present work.

$$\begin{bmatrix} a_{pred} \\ b_{pred} \\ c_{pred} \end{bmatrix} = \begin{bmatrix} \frac{a_{net}}{\lambda_a} \\ \frac{b_{net}}{\lambda_b} \\ \frac{c_{net}}{\lambda_c} \end{bmatrix} \tag{7}$$

*2.3. Training*

Before training, we know the true values of $\mu_x$ and $\mu_y$ at each step in the data set; when the training coefficients are $a_{net}$, $b_{net}$, and $c_{net}$, the true yaw rate coefficients $a_t$, $b_t$, and $c_t$ are obtained by calculating the true values of $\mu_x$ and $\mu_y$ at every step. Corresponding to each step, there is a set of $\mu_x$ and $\mu_y$, which contains the next $n$ (number of steps in the neural network) steps $\mu_x$ and $\mu_y$. Through this set, combined with (1), the true values of $a_t$ and $b_t$ can be obtained by the polynomial regression [26] fitting function and the true value of $c_t$ can be obtained by the linear regression [27] fitting function.

Each coefficient has a different value range; both the true value and the mean square error loss of the coefficient are very small. Therefore, it is necessary to normalize coefficients during training [28]. For normalization, when $a_t$, $b_t$, and $c_t$ are used for training, they are weighted with $\lambda_a = 1000$, $\lambda_b = 10{,}000$, and $\lambda_c = 1$, respectively. In order to minimize the loss, the $a_{net}$, $b_{net}$, and $c_{net}$ training targets are the values obtained after multiplying the true coefficients $a_t$, $b_t$, and $c_t$ by $\lambda_a$, $\lambda_b$, and $\lambda_c$, respectively. The mean square error loss of $a_{net}$, $b_{net}$, and $c_{net}$ for the training is as follows:

$$L_{mse} = (\lambda_a a_t - a_{net})^2 + (\lambda_b b_t - b_{net})^2 + (\lambda_c c_t - c_{net})^2 \tag{8}$$

The optimizer uses the loss function (8) to optimize the weight and bias of the network model in the direction of minimizing $L_{mse}$. In the training process, $W_a$, $W_b$, $W_c$, $b_a$, $b_b$, and $b_c$ are trained according to the coefficients of the yaw rate. The size of each training batch is 12,000, the learning rate is 0.1, and the number of iterations in training is 10,000. Adam's optimizer [29] was used to handle the weights and biases, and the optimizer parameters were the following: $\beta_1 = 0.9$, $\beta_2 = 0.999$, and $\varepsilon = 10^{-8}$.

The driving data of an NTV prototype [3] are used for training, as shown in Figure 3. The NTV prototype has two steering front wheels and one driving rear wheel. A global navigation satellite system/inertial navigation system (GNSS/INS, Shanghai, China) and wheel speed Sensor (WSS, Zhejiang, China) were installed on the prototype to provide real-time data as the input to the network model. The $a_y$ in the input is directly obtained by the inertial measurement unit (IMU) embedded in the GNSS/INS. $v_x$ and $v_y$ were calculated from the data obtained by IMU and WSS. The coefficients $a$, $b$, and $c$ can be obtained by calculating the true values of $\mu_x$ and $\mu_y$ at every step, which can be obtained by GNSS/INS.

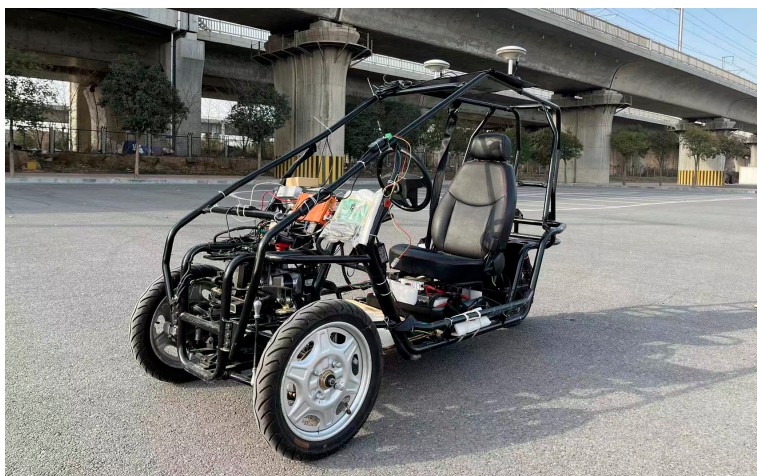

**Figure 3.** The prototype of an NTV.

The model of GNSS/INS is RoHS-X2-CAN; the WSS model is TE-ABS-181; the vehicle chip model of the vehicle control unit (VCU) is NXP5744P; the network model is calculated by a Jetson Xavier NX; all driving data were recorded by CAN-DTU200; the above devices communicate with each other by CAN bus [30]. Through the prototype test, 100 datasets were obtained for training the network model. Each dataset contained 120 s of driving data (12,000 time-sequential data points). In addition to the training set, 10 additional datasets were used as the test sets to obtain the final results.

For performance comparison, the number of hidden layers (*Nol*), the number of hidden layers nodes (*Non*), and the number of steps (*n*) of the network model were different. The test sets were used to find the appropriate network structure and the number of steps. After training, the root mean squared error (RMSE) of the test sets was calculated, and the operation times of each step are shown in Figure 4. The operation time was determined by a Jetson Xavier NX test.

| *Nol/Non* \ *n* | 10 | 15 | 20 | 25 | 30 |
|---|---|---|---|---|---|
| 1/6 | 0.1268 | 0.1121 | 0.1083 | 0.0927 | 0.1172 |
| 1/12 | 0.1105 | 0.1004 | 0.0922 | 0.0867 | 0.0901 |
| 1/18 | 0.1001 | 0.0912 | 0.0856 | 0.0799 | 0.0812 |
| 2/6 | 0.0993 | 0.0862 | 0.0701 | 0.0691 | 0.0815 |
| 2/12 | 0.0859 | 0.0712 | 0.0639 | 0.0631 | 0.0723 |
| 2/18 | 0.0761 | 0.0701 | 0.0637 | 0.0640 | 0.0695 |
| 3/6 | 0.0712 | 0.0699 | 0.0598 | 0.0598 | 0.0623 |
| 3/12 | 0.0698 | 0.0649 | 0.0526 | 0.0571 | 0.0602 |
| 3/18 | 0.0672 | 0.0631 | 0.0523 | 0.0501 | 0.0532 |

(**a**)

| *Nol/Non* \ *n* | 10 | 15 | 20 | 25 | 30 |
|---|---|---|---|---|---|
| 1/6 | 2 | 3 | 5 | 7 | 9 |
| 1/12 | 2 | 4 | 5 | 8 | 10 |
| 1/18 | 3 | 5 | 7 | 9 | 11 |
| 2/6 | 3 | 5 | 6 | 9 | 11 |
| 2/12 | 4 | 6 | 7 | 9 | 14 |
| 2/18 | 5 | 7 | 9 | 12 | 15 |
| 3/6 | 5 | 8 | 10 | 12 | 14 |
| 3/12 | 6 | 9 | 12 | 14 | 15 |
| 3/18 | 7 | 10 | 13 | 15 | 17 |

(**b**)

**Figure 4.** The result of training. (**a**) RMSE, Unit: °/s; (**b**) The operation time of each step, Unit: ms.

Since the input sends data at a time interval of 0.01 s, the network model should not spend more than 10 ms in performing each step calculation. To balance the operation time and performance of the network model, a 20-step network structure with 2 hidden layers and 12 hidden layer nodes in each layer was chosen. The final size of the network model used for burning was 7072 kB. The network model under this structure was applied to the test sets, and the prediction result is shown in Figure 5.

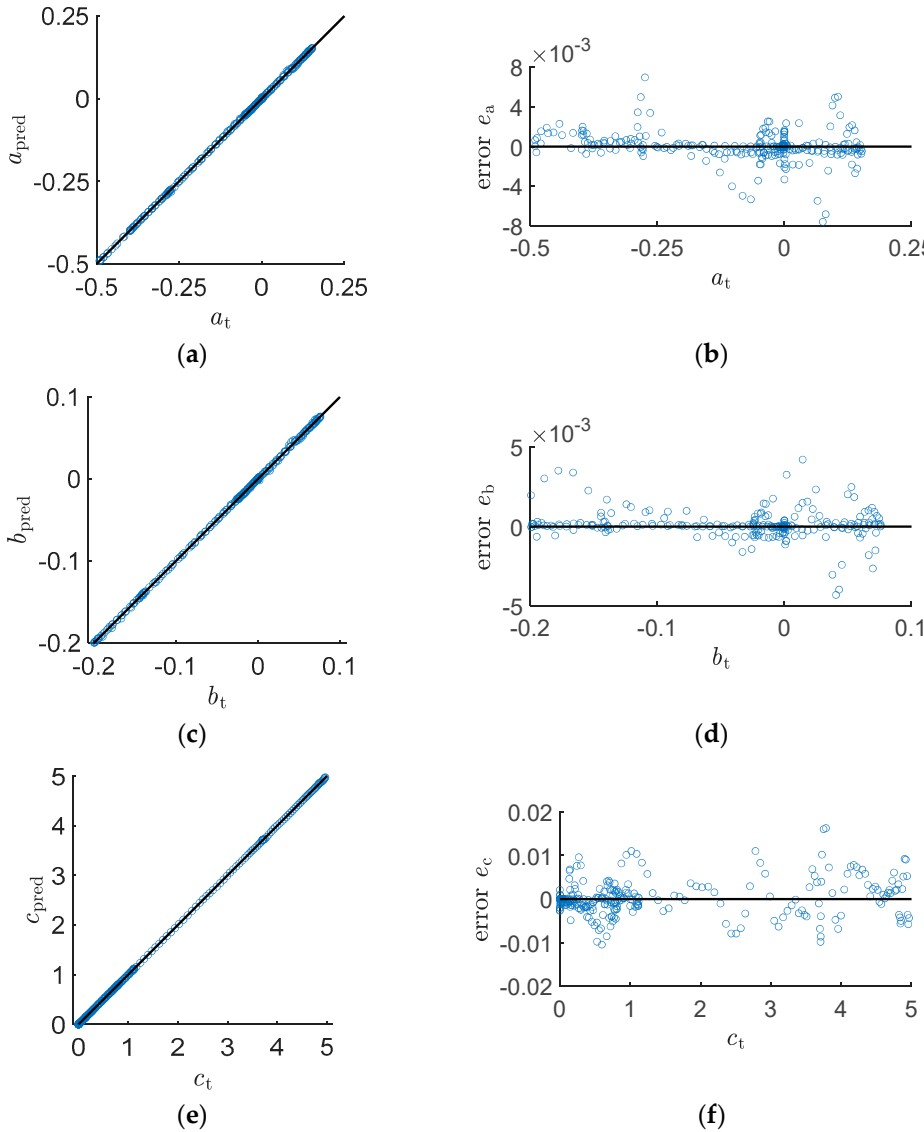

**Figure 5.** Predictive results and errors of the network model: (**a**) $a_t$ and $a_{pred}$; (**b**) the error of $a_{pred}$; (**c**) $b_t$ and $b_{pred}$; (**d**) the error of $b_{pred}$; (**e**) $c_t$ and $c_{pred}$; (**f**) the error of $c_{pred}$.

The yaw rate and its error calculated from the prediction results are shown in Figure 6. $\varphi_t$ was obtained by calculating the true yaw rate coefficients $a_t$, $b_t$, and $c_t$.

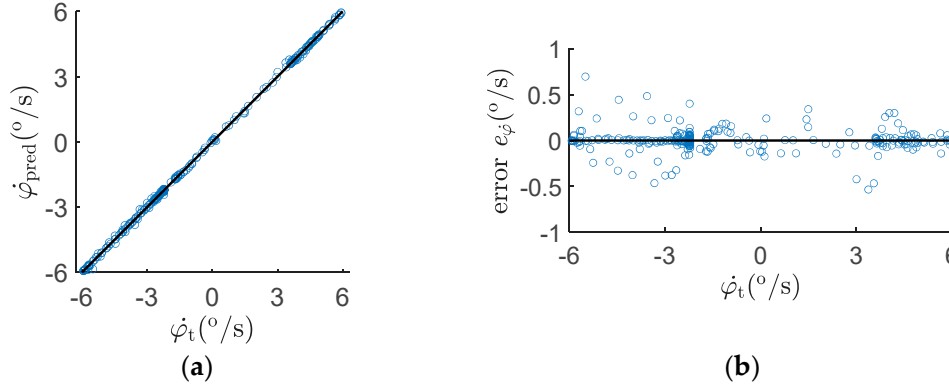

**Figure 6.** Predictive results $\varphi_t$ and $\varphi_{pred}$ and the error of $\varphi_{pred}$: (**a**) $\varphi_t$ and $\varphi_{pred}$; (**b**) the error of $\varphi_{pred}$.

The results show that the maximum absolute error of the yaw rate predicted by the network model was 0.3468 deg/s. The mean absolute error was 0.0013 deg/s. The relative error at the maximum absolute error was 6.31%. The average relative error was 4.07%.

The value range of the yaw rate in the test set was between −6 deg/s and 6 deg/s, and the maximum absolute error appeared near −6 deg/s. The maximum absolute error of the yaw rate was 0.2008 deg/s in an interval between −2.5 deg/s and −2.5 deg/s. The mean absolute error was 0.000264 deg/s.

These errors mainly come from the three coefficients of the network model output. The accuracy of the output with the network model is analyzed in the discussion section of this work.

### 3. Tilting Feedforward Synchronous Control

In this section, we propose that the TFSC is synchronized with the active tilting motion and yaw motion of NTVs. The TFSC is described based on the NTV prototype above.

When the prototype is driving forward, the prototype takes the ideal tilting angle [9,17] as the target tilting angle, which is calculated by (9) [9,17]. The ideal tilting angle and the yaw rate are synchronized in the time domain.

$$\theta_t = \frac{v\dot{\varphi}}{g} \tag{9}$$

where $\theta_t$ is the target tilting angle; $\dot{\varphi}$ is the yaw rate; $v$ is the driving velocity; and $g$ is the acceleration of gravity.

In the traditional direct tilt control (DTC), shown in Figure 7, the prototype takes (9) as the control target to control the tilting angle.

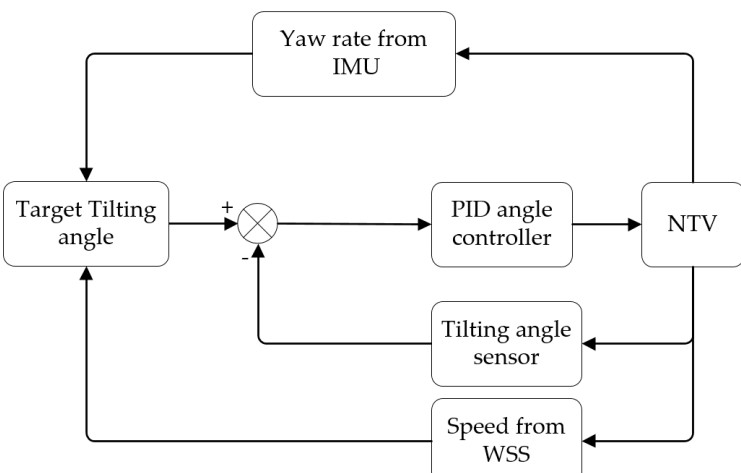

**Figure 7.** The diagram of DTC.

Driving at the driving velocity $v$ = 30 km/h, the yaw rate, target tilting angle, and true tilting angle of the NTV prototype are shown in Figure 8.

There is a time lag between the true and target tilting angles. The average lag time $t_b$ = 0.17 s.

To synchronize the true tilting angle and the yaw rate in the time domain, the TFSC method causes the control target of the tilting angle at time $t_0$ to be calculated as follows:

$$\theta_{t0} = \frac{v_{t0}\dot{\varphi}_{t0+tb}}{g} \tag{10}$$

where $v_{t0}$ is the driving velocity at time $t_0$; $\theta_{t0}$ is the target tilting angle at time $t_0$; and $\dot{\varphi}_{t0+tb}$ is the yaw rate at time $t_0+t_b$, where $\dot{\varphi}_{t0+tb}$ is obtained by the prediction of the network model.

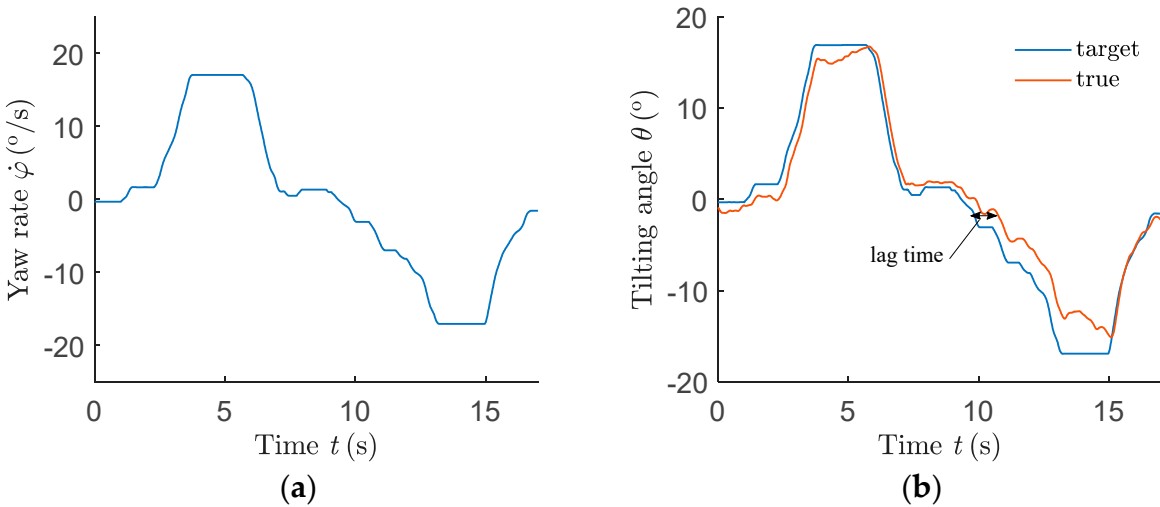

**Figure 8.** Lag time of true tilt angle with DTC: (**a**) yaw rate; (**b**) tilting angle.

In the TFSC, as shown in Figure 9, the predicted yaw rate output from the network model is used as the basis for calculating the control target. The target tilting angle of synchronous control is earlier than the yaw rate in the time sequence, which is used to compensate for the time lag of the true tilting angle caused by suspension damping [30,31]. The network model is used to realize data feedforward in order to realize the synchronization of the active tilting motion and the yaw motion. The NTV prototype takes (10) as the control target to control the tilting angle, and the synchronization effect achieved is verified in the experiments section of this work.

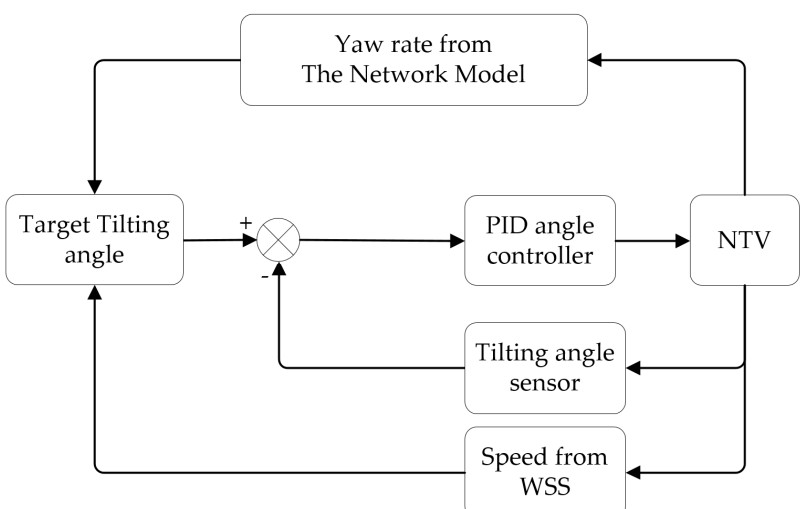

**Figure 9.** The diagram of TFSC.

## 4. Experiments

The above prototype was used in experiments as a test NTV to acquire the dataset, as shown in Figure 10. Four typical scenarios [32] are provided to analyze the performance of the proposed TFSC and prove its applicability. The proposed method can be expanded to other scenarios with structured roads easily. The illustrative examples are tested by the experiments.

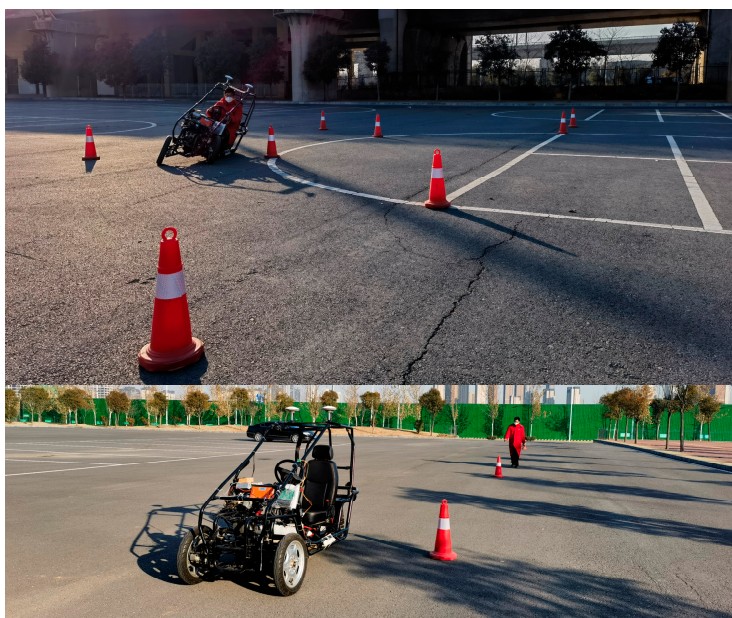

**Figure 10.** The experimental process.

When obtaining the training set and test set, the prototype experiment was carried out on the "S"-type route with a pile spacing of 10 m and the "C"-type route with a turning radius of 5.6 m. The "C"-type route involves only one turning process, while the "S"-type route involves many turning processes. Two driving scenarios were considered: in scenario (1), the prototype was driven at a constant velocity (30 km/h); and in scenario (2), the prototype was driven within the prescribed route at a velocity that was entirely controlled by the driver.

To verify the effect of the TFSC method based on prediction, the trained network model was downloaded onto the VCU and Jetson Xavier NX of the NTV prototype, and the NTV prototype was driven on the following four routes, as shown in Figure 11. The four types of routes are the "S"-type route, the "C"-type route, the single lane change route, and the double lane change route. When the TFSC was adopted, the $t_b$ in (10) was 0.17 s.

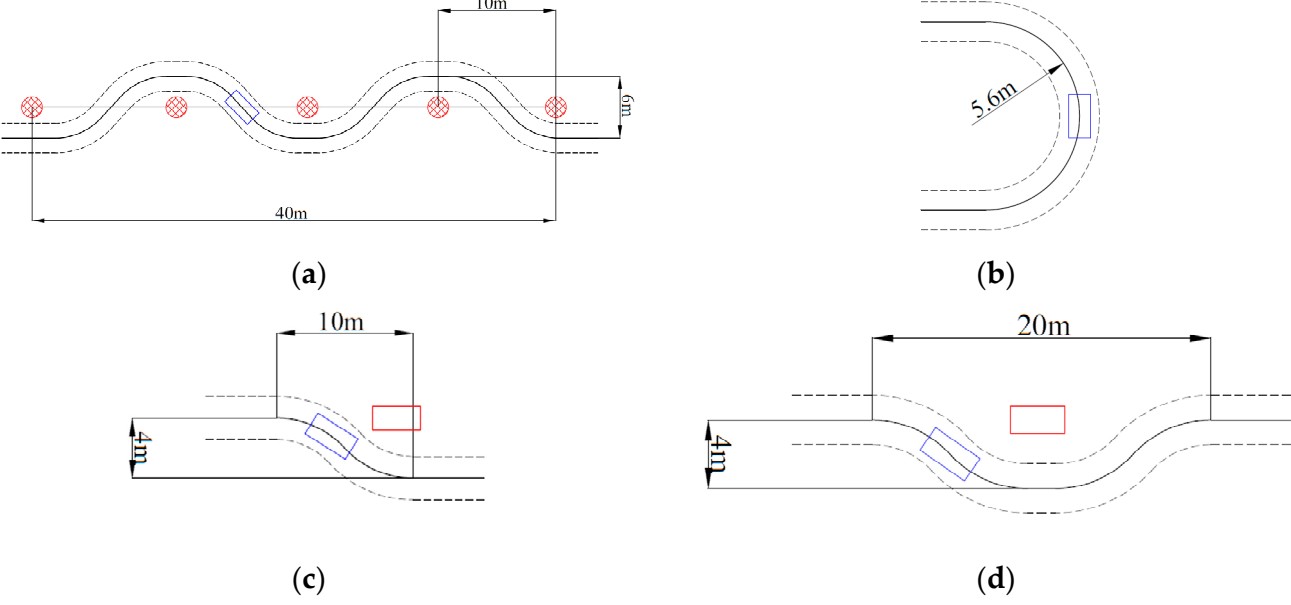

**Figure 11.** The route of the validation experiment: (**a**) the "S"-type route; (**b**) the "C"-type route; (**c**) the single lane change route; (**d**) the double lane change route.

### 4.1. "S"-Type Route Experiment

The data of the NTV prototype when driving on the "S"-type route are shown in Figure 12. The true tilting angles of the NTV prototype with DTC and TFSC are compared with the ideal tilting angle. The ideal tilting angle is synchronized with the yaw rate [9].

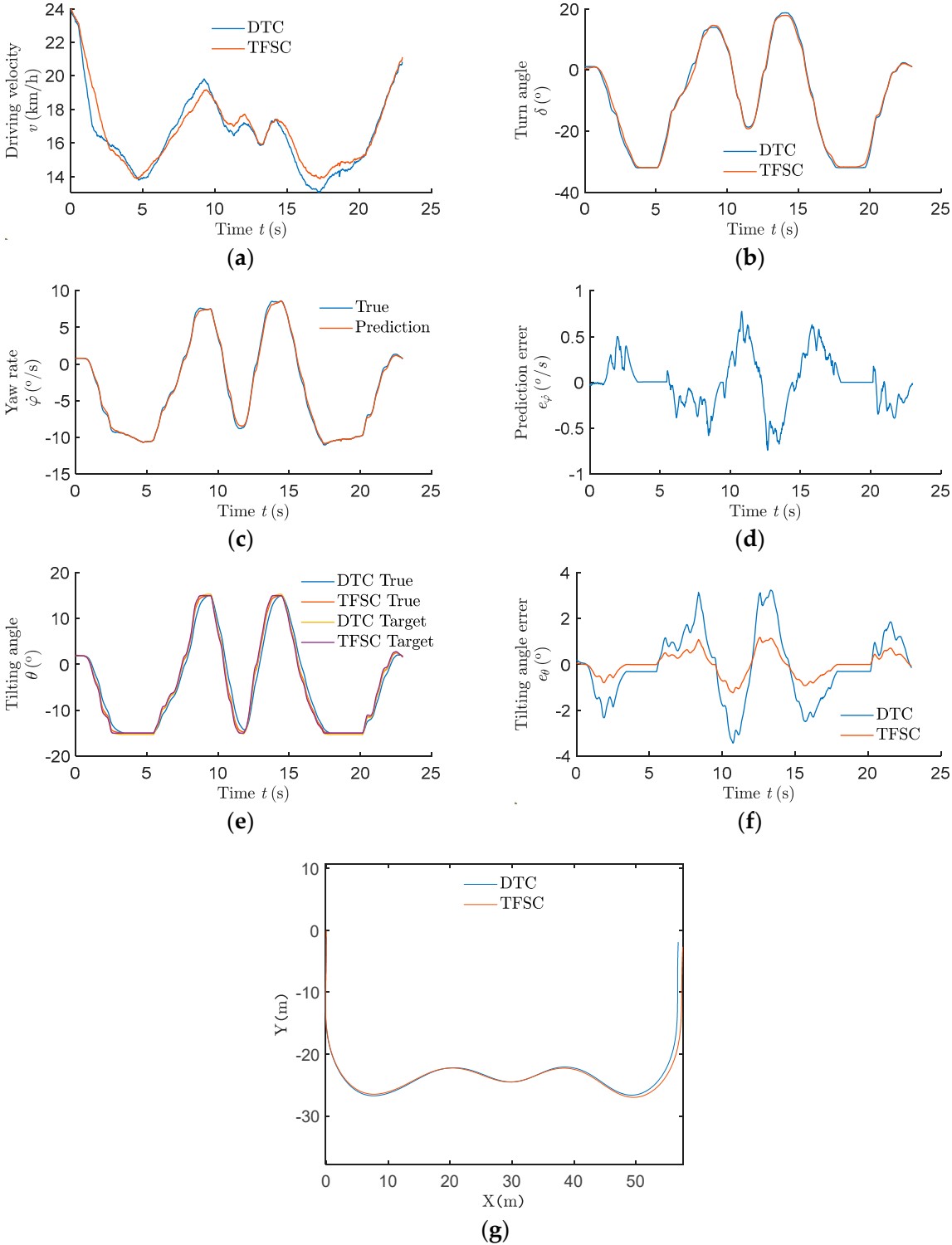

**Figure 12.** The experiment results of the "S"-type route. (**a**) driver input driving velocity; (**b**) driver input turn angle; (**c**) yaw rate; (**d**) prediction error of $\dot{\varphi}$; (**e**) tilting angle; (**f**) the error of tilting angle; (**g**) the trajectory.

When the prototype drove on the "S"-type route, the maximum absolute error of the prediction was 0.7531°/s, and the average absolute error was 0.2026°/s. The true tilting angle with TFSC was closer to the corresponding target [9]. The tilting angle error is shown in Table 1.

**Table 1.** The tilting angle error of the "S"-type route experiment.

|  | Maximum Absolute Error (°) | Average Absolute Error (°) | Average Lag Time (s) |
|---|---|---|---|
| DTC | 3.317 | 1.179 | 0.21 |
| TFSC | 1.208 | 0.401 | 0.11 |

*4.2. "C"-Type Route Experiment*

The data of the NTV prototype when driving on the "C"-type route is shown in Figure 13. The true tilting angles of the NTV prototype with DTC and TFSC are compared with the ideal tilting angle.

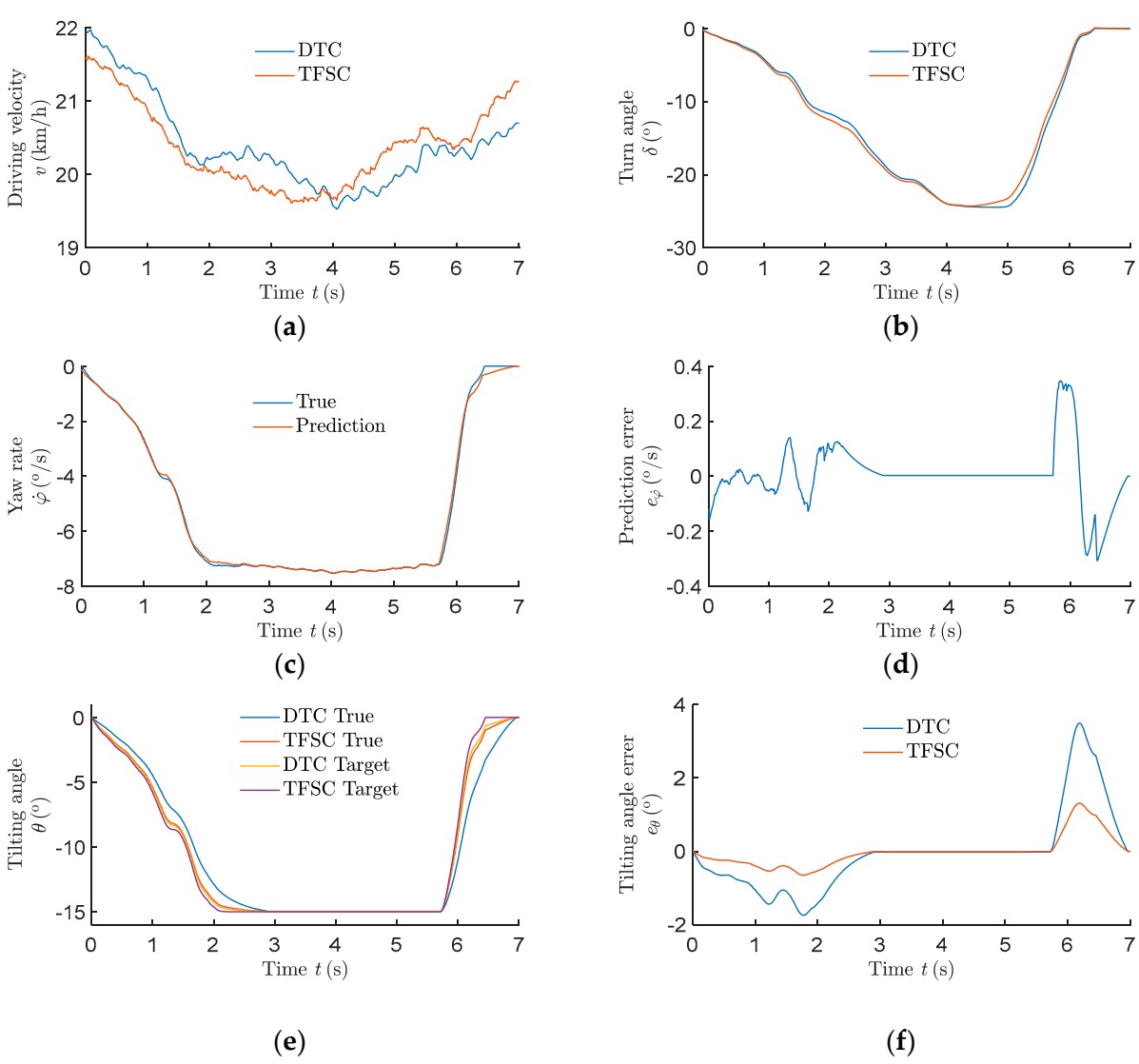

**Figure 13.** *Cont.*

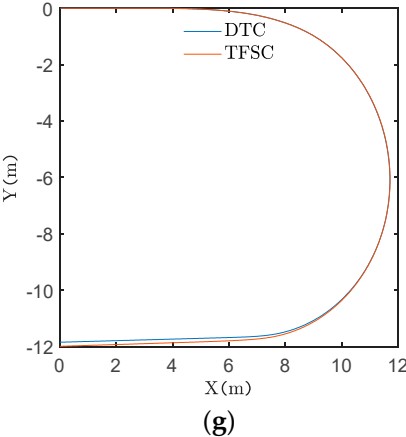

**(g)**

**Figure 13.** The experiment results of the "C"-type route: (**a**) driver input driving velocity; (**b**) driver input turn angle; (**c**) yaw rate; (**d**) prediction error of $\dot{\varphi}$; (**e**) tilting angle; (**f**) the error of tilting angle; (**g**) the trajectory.

When the prototype drove on the "C"-type route, the maximum absolute error of the prediction was $0.3466°/s$, and the average absolute error was $0.0445°/s$. The true tilting angle with TFSC was closer to the corresponding target [9]. The tilting angle error is shown in Table 2.

**Table 2.** The tilting angle error of the "C"-type route experiment.

|       | Maximum Absolute Error (°) | Average Absolute Error (°) | Average Lag Time (s) |
|-------|-----------------------------|-----------------------------|-----------------------|
| DTC   | 3.488                       | 0.5167                      | 0.07                  |
| TFSC  | 1.308                       | 0.1938                      | 0.03                  |

### 4.3. Single Lane Change Route Experiment

The data of the NTV prototype when driving in the single lane change route are shown in Figure 14. The true tilting angles of the NTV prototype with DTC and TFSC are compared with the ideal tilting angle.

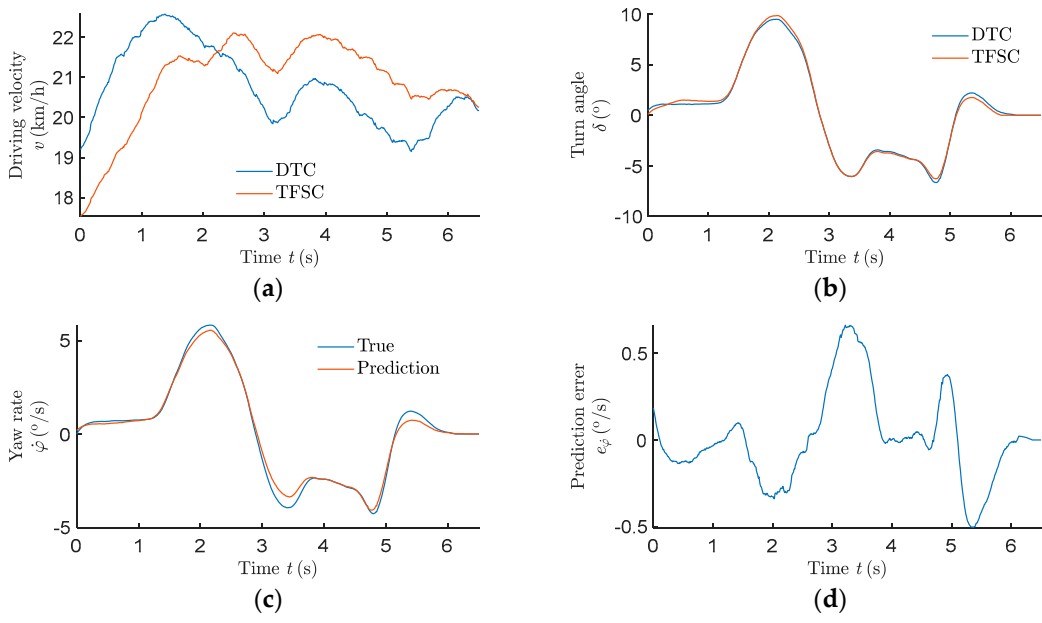

**Figure 14.** *Cont*.

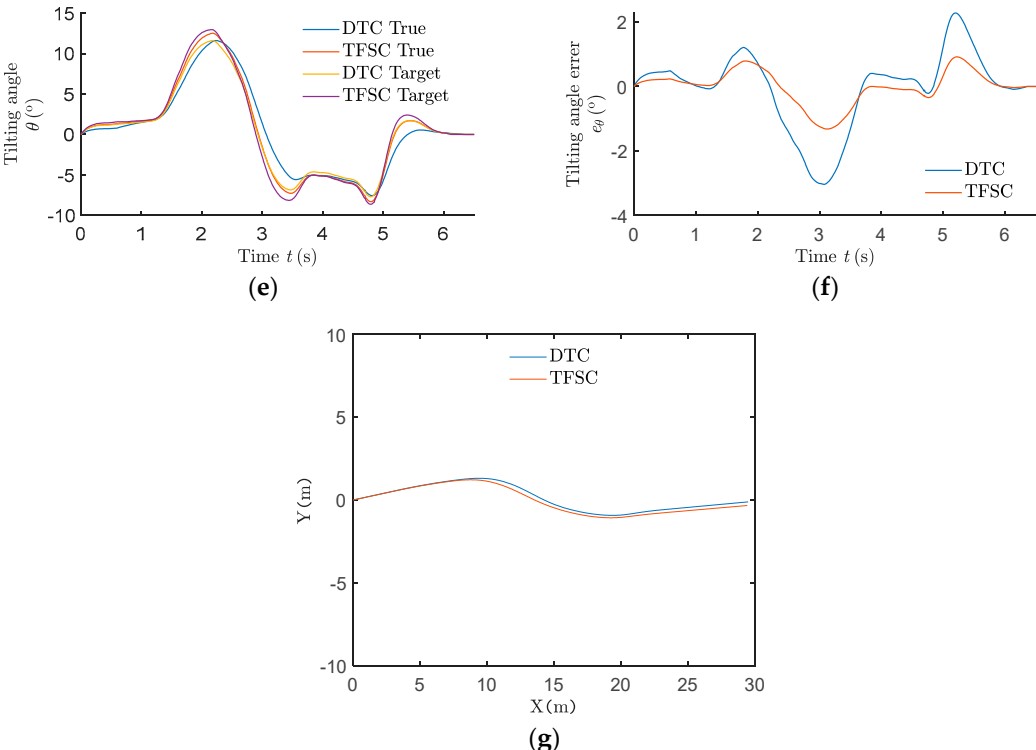

(e)

(f)

(g)

**Figure 14.** The experiment results of the single lane change route: (**a**) driver input driving velocity; (**b**) driver input turn angle; (**c**) yaw rate; (**d**) prediction error of $\dot{\varphi}$; (**e**) tilting angle; (**f**) the error of tilting angle; (**g**) the trajectory.

When the prototype drove on the single lane change route, the maximum absolute error of the prediction was $0.6537°/s$, and the average absolute error was $0.1335°/s$. The true tilting angle with TFSC was closer to the corresponding target [9]. The tilting angle error is shown in Table 3.

**Table 3.** The tilting angle error of the single lane change route experiment.

|  | Maximum Absolute Error (°) | Average Absolute Error (°) | Average Lag Time (s) |
|---|---|---|---|
| DTC | 3.029 | 0.5963 | 0.16 |
| TFSC | 1.321 | 0.2716 | 0.07 |

*4.4. Double Lane Change Route Experiment*

The data of the NTV prototype when driving in the double lane change route are shown in Figure 15. The true tilting angles of the NTV prototype with DTC and TFSC are compared with the ideal tilting angle.

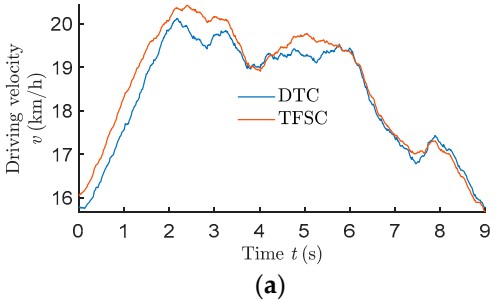

(a)

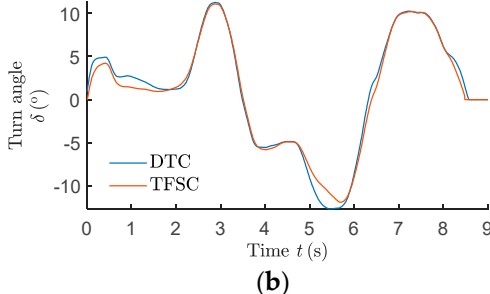

(b)

**Figure 15.** *Cont.*

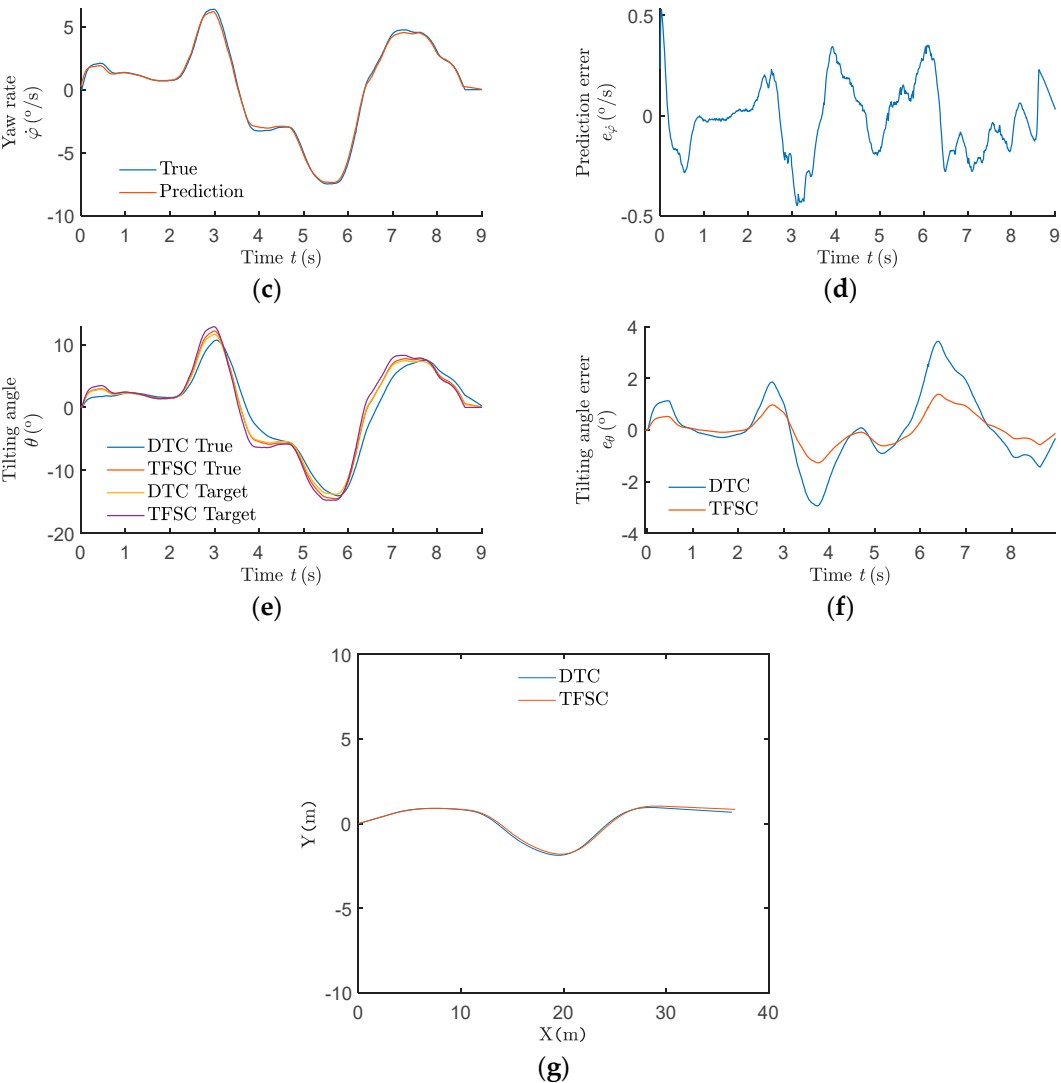

**Figure 15.** The experiment results of the double lane change route: (**a**) driver input driving velocity; (**b**) driver input turn angle; (**c**) yaw rate; (**d**) prediction error of $\dot{\varphi}$; (**e**) tilting angle; (**f**) the error of tilting angle; (**g**) the trajectory.

When the prototype drove in the double lane change route, the maximum absolute error of the prediction was $0.4492°/s$, and the average absolute error was $0.1023°/s$. The true tilting angle with TFSC was closer to the corresponding target [9]. The tilting angle error is shown in Table 4.

**Table 4.** The tilting angle error of the double lane change route experiment.

|  | Maximum Absolute Error (°) | Average Absolute Error (°) | Average Lag Time (s) |
|---|---|---|---|
| DTC | 3.409 | 0.7089 | 0.18 |
| TFSC | 1.375 | 0.3229 | 0.10 |

### 4.5. Analysis of Experiment Results

According to Tables 1–4, the tilting motion performance under the control of TFSC was better than that of traditional DTC. The lag time in Tables 1–4 can meet the performance requirements of vehicle suspension [33]. In the four typical scenarios shown in Sections 4.1–4.4, TFSC reduces the average absolute error of the tilting angle by 65.9%, 63.8%, 54.7%, and 54.5%, respectively; TFSC reduces the average lag time of the tilting

angle by 47.6%, 57.1%, 56.2%, and 44.4%, respectively. The tilting motion performances (e.g., average lag time, maximum absolute error and average absolute error of tilting angle) of TFSC under four scenarios were compared, and the results are shown in Figure 16.

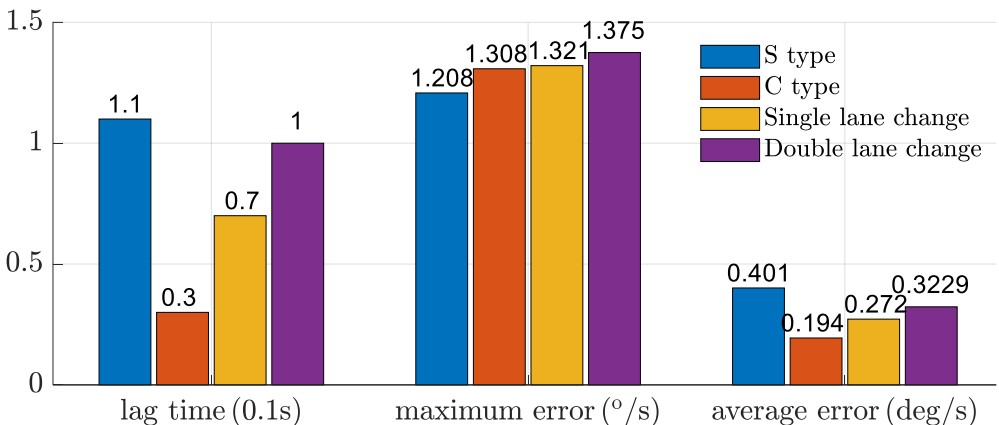

**Figure 16.** Analysis results.

In terms of error, the performance of TFSC in the four scenarios was relatively average; in terms of reducing lag time, TFSC performed best in the "C"-type route.

## 5. Discussion

By comprehensively analyzing the contents of Figures 4–6, it can be seen that the calculation speed of the prediction algorithm is fast enough, but there are also certain errors. The main reason for prediction error could be that the output of the three coefficients $a$, $b$, and $c$ are inaccurate. The error of coefficient $c$ is relatively large, but the error of the yaw rate finally calculated by the three coefficients is relatively small, indicating that the influence of coefficient $c$ on the yaw rate in the prediction process is relatively less than the other two coefficients.

According to Figure 6, for the RNN method, no matter how much longer the input "memory" is compared to others, the prediction effect is not necessarily better. When using the RNN method, finding the appropriate "memory" length is necessary.

When designing the prediction algorithm of the yaw rate, we assumed that the lateral acceleration and the longitudinal velocity between each step were constant. This will certainly lead to some prediction error, but from the prediction results, this error is very small and can be applied to the proposed TFSC.

The NTV's tilting motion performance in the proposed TFSC in the "C"-type route and the single lane change route with only one steering process is better than that in the "S"-type route and the double lane change route with multiple steering processes, as shown in Figure 16. There are two reasons for this situation: (1) the accuracy of the predicted yaw rate; (2) the $t_b$ in the TFSC is currently a constant that does not change with the operating scenarios.

Regarding the accuracy of the predicted yaw rate, the value range of the yaw rate in the test set was between $-6$ deg/s and 6 deg/s, and the maximum absolute error appeared near $-6$ deg/s. The maximum absolute error of the yaw rate was 0.2008 deg/s in a common working scenario (between $-2.5$ deg/s and $-2.5$ deg/s). The mean absolute error was 0.000264 deg/s.

The $t_b$ is a variable in the steering process, as shown in Figure 8. To calculate the target tilting angle, the $t_b$ is approximated to a constant in the TFSC. If $t_b$ can be predicted by deep learning or other methods, the tilting motion performance of TFSC could be further improved. Future works should aim to address this possibility.

## 6. Conclusions

(1) This work researched the calculating method for the prediction of the continuous yaw rate of NTVs in the time domain. A network model was designed based on an RNN to predict the NTV yaw rate. The network model spends less than 10 ms performing each step calculation. The root mean squared error of prediction is $0.0639°/s$.

(2) The tilting feedforward synchronous control (TFSC) method for NTVs, based on the predicted value of the yaw, rate is proposed. This method reduces the maximum average error of the tilting angle by 54.5%; the average lag time of the tilting angle is reduced by 44.4%. The experiment showed that this method can effectively synchronize the tilting motion of the NTV with the yaw motion.

(3) All sensor data used in the network model and the TFSC were obtained through a prototype and its onboard sensors. The proposed TFSC can be downloaded onto the onboard chip for use, which shows the practicability of the TFSC.

## 7. Patents

There are patents resulting from the work reported in this manuscript. The patent numbers of these are CN115214619A, CN115214620A, and CN115257706A.

**Author Contributions:** Conceptualization, R.G. and Y.W.; methodology, R.G. and H.L.; software, R.G. and X.Z.; validation, R.G. and Y.W.; writing—original draft preparation, R.G.; writing—review and editing, H.L.; supervision, Y.W., W.W. and S.X.; project administration, W.W., R.G. and N.L. All authors have read and agreed to the published version of the manuscript.

**Funding:** This research was funded by the special construction project of "double first-class" scientific research of China (grant number: 2022AC025). This work was supported by the Government Procurement Project of China (grant number: CLF0121SZ08QY08P).

**Data Availability Statement:** The experiment data and software of the mathematical model can be found at https://pan.baidu.com/s/12VmE0m_uK-qHnsxP3LIA-A (accessed on 16 February 2023). The extraction code of the downloadable resources is 1111.

**Acknowledgments:** Authors thank Zuoqi Technology Co., Ltd. for providing the facilities and the site of the experiments.

**Conflicts of Interest:** The authors declare no conflict of interest. The funders had no role in the design of the study; in the collection, analyses, or interpretation of data; in the writing of the manuscript; or in the decision to publish the results.

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
