# Peer review of "Yaw Rate Prediction and Tilting Feedforward Synchronous Control of Narrow Tilting Vehicle Based on RNN"

_machines, doi:10.3390/machines11030370_

Round 1
Reviewer 1 Report
The article named Yaw Rate Prediction and Tilting Feedforward Synchronous
Control of Narrow Tilting Vehicle Based on Recurrent Neural network (RNN) is an interesting study presenting a new approach to the tilting feedforward synchronous control for Narrow Tilting Vehicles.
The study is logically divided into chapters that deal with the use of RNN in the process of vehicle trajectory prediction including the methodology of the network training and applying the results to the TFSC controller.
I really appreciate the experimental part of the article describing the test drives with the NTV and also the comparison of the DTC and TFSC methods results clearly presented in the chapter 4.
The results of the work are then discussed pointing out possible problems with the selected algorithm and the conclusion is supported by the study results, which are also the subject of several patents.
Thus, I recommend to accept the article in present form for publishing.
Author Response
Response to Reviewer 1 Comments
Dear reviewer:
Thank you very much for your letter dated February 20th, in which you sent us the Referees’ Reports on our manuscript. Thankyou very much for your time and efforts in considering this manuscript. Thank you for your comments and affirmations, these help us a lot.
The authors once again thank you for your careful review of the manuscript in your busy schedules.
Sincerely yours
Ruolin Gao, Haitao Li, Ya Wang, Shaobing Xu, Wenjun Wei, Xiao Zhang, and Na Li
Reviewer 2 Report
This paper proposed the tilting feedforward synchronous control for NTV, where RNN proposes the yaw
rate. The topic is deserved to explore. However, I have some concerns about this study.
1. (1) is not consistent with equations in Figure .1. Please clarify.
2. How could the author obtain the true values of a?, b?, and c?? If there is any analytical way to calculate
these values, then what is the motivation for RNN?
3. Could the author give more details about (2)? That is, the definition of yaw angle should be based on
the current vehicle body frame or global frame. Also, how to obtain (2) from the geometry view?
4. It is also recommended that the author give the tracking trajectory of each method in the experiment
results
Author Response
Response to Reviewer 2 Comments
Dear reviewer:
The authors are very grateful for your comments on the manuscript. According to your advice, The authors amend the relevant part of the manuscript. The question raised are responded as follows
Point 1: (1) is not consistent with equations in Figure .1. Please clarify.
Response 1: Thanks for the reminder, the authors noticed the error in (1) and revised it.
Point 2: How could the author obtain the true values of a?, b?, and c?? If there is any analytical way to calculate these values, then what is the motivation for RNN?
Response 2: The author added how to get the true values of a?, b?, and c? in lines 169-174. The details are as follows.
“When training coefficients anet, bnet, and cnet, the true yaw rate coefficients at, bt, and ct are obtained by calculating the true values of μx and μy at every step. Before training, since we know the true values of μx and μy at each step in the data set. Corresponding to each step, there is a set of μx and μy, which contains the next n (Number of steps in the neural network) steps μx and μy. Through this set, combined with (1), the true values of at and bt can be obtained by the polynomial regression [26] fitting function and the true value of ct can be obtained by the linear regression [27] fitting function.”
The reasons for using RNN method are as follows.
For a vehicle with a known trajectory, it is not difficult to obtain its real-time a?, b?, and c? value and yaw rate. However, in the control of NTV, in order to realize the synchronous response of the NTV tilting motion and the NTV yaw motion, we need to know the future continuous yaw rate in advance to do feed-forward for the control of the tilting motion. Therefore, in this manuscript, the authors predict the three coefficients through RNN, so as to realize the prediction of continuous yaw rate.
Point 3: Could the author give more details about (2)? That is, the definition of yaw angle should be based on. the current vehicle body frame or global frame. Also, how to obtain (2) from the geometry view?
Response 3: The authors agree with the comment made by the reviewer and made supplementary explanations in lines 129-133. The details are as follows.
“Relative to the NTV coordinate at time t0, the yaw angle φ at time t can be expressed as the angle between the x-axis of NTV at time t and the x-axis of NTV at time t0, that is, the angle between the trajectory tangent at time t and the x-axis of NTV at time t0.”
At the same time, in order to better represent this principle,the authors supplemented φ in Figure 2.
Point 4: It is also recommended that the author give the tracking trajectory of each method in the experiment results.
Response 4: The reviewer provided valuable comments, which led the authors to improve the manuscript. The authors supplemented the trajectory of the prototype during the experiments in the manuscript.
The authors once again thank you for your careful review of the manuscript in your busy schedules. The relevant statements in the manuscript have been revised. The co-authors each have proofread the manuscript to make every effort to ensure that the revised manuscript is more readable. The authors hope the paper in its revised version is acceptable.
Sincerely yours
Ruolin Gao, Haitao Li, Ya Wang, Shaobing Xu, Wenjun Wei, Xiao Zhang, and Na Li